# Distill DSM: Computationally efficient method for segmentation of medical imaging volumes

**Harsh Maheshwari**[1]                                    HARSHMAHESHWARI135@GMAIL.COM
**Vidit Goel**[1]                                                        GVIDIT98@GMAIL.COM
**Ramanathan Sethuraman**[2]                RAMANATHAN.SETHURAMAN@INTEL.COM
**Debdoot Sheet**[1]                                        DEBDOOT@EE.IITKGP.AC.IN
[1] *Indian Institute of Technology, Kharagpur*
[2] *Intel Technology India Pvt. Ltd, Bangalore*

**Editors:** Under Review for MIDL 2021

## Abstract

Accurate segmentation of volumetric scans like MRI and CT scans is highly demanded for surgery planning in clinical practice, quantitative analysis, and identification of disease. However, accurate segmentation is challenging because of the irregular shape of given organ and large variation in appearances across the slices. In such problems, 3D features are desired in nature which can be extracted using 3D convolutional neural network (CNN). However, 3D CNN is compute and memory intensive to implement due to large number of parameters and can easily over fit, especially in medical imaging where training data is limited. In order to address these problems, we propose a distillation-based depth shift module (Distill DSM). It is designed to enable 2D convolutions to make use of information from neighbouring frames more efficiently. Specifically, in each layer of the network, Distill DSM learns to extract information from a part of the channels and shares it with neighbouring slices, thus facilitating information exchange among neighbouring slices. This approach can be incorporated with any 2D CNN model to enable it to use information across the slices with introducing very few extra learn-able parameters. We have evaluated our model on BRATS 2020, heart, hippocampus, pancreas and prostate dataset. Our model achieves better performance than 3D CNN for heart and prostate datasets and comparable performance on BRATS 2020, pancreas and hippocampus dataset with simply 28% of parameters compared to 3D CNN model.

**Keywords:** Deep learning, volumetric segmentation, parameter efficient 3D CNN, distillation, channel shifting

## 1. Introduction

Medical imaging using Computed Tomography(CT) and Magnetic Resonance Imaging (MRI) is frequently used in clinical practice for investigating a wide range of conditions, e.g., injury prediction, disease diagnosis, surgery simulation, therapeutic planning, etc. It is often required to segment the portion of interest in a given CT or MR to interpret or analyse the clinical conditions.

Manual segmentation of medical images is tedious and labour intensive work and often leads to high variation across reporters, which motivates the need to automate the segmentation process. With the advances in deep learning methods, convolutional neural network (CNN) are becoming increasingly popular in being applied to various medical image segmentation tasks to increase consistency across multiple human experts.

Medical images like CT and MRI are 3D in nature and widely used in clinical diagnosis. In order to perform the segmentation of volumetric data, we can employ the following possible strategies. The first is by considering the 3D volume as set of individual 2D slices and training 2D CNN for segmenting the structures of interest in 2D slices. Another approach is to enable the network operations to process volumetric data by using 3D CNN and train the 3D CNN for volumetric segmentation. Using 2D CNNs for segmentation results in a computationally light model with faster inference time. However, it does not take into account the information from adjacent slices, resulting in a model with lowered segmentation accuracy. On the other hand, 3D CNN is able to incorporate information from adjacent slices for better quality of segmentation and has the same spatial field of view as that in a 2D CNN, but it requires higher computation cost resulting in lowered throughput and higher latency. On account of the large number of parameters, 3D CNNs are prone to overfitting, especially with small dataset.

In order to bridge the performance gap between 2D CNN and 3D CNN, we propose a simple and computationally efficient technique with computational complexity in the order of 2D CNN, while being able to incorporate the interslice information for enhanced quality. In this paper, we introduce a novel component termed Distill DSM, which is able to effectively model information along the depth dimension, motivated by TSM[1] (Lin et al., 2019) originally for action recognition in videos. The proposed module can be inserted in any 2D segmentation network to improve its performance with a negligible increase in the number of parameters and order of computation. In each layer of the network where present, Distill DSM learns to extract information that is useful for the current slice and information that is useful for the immediate neighbours thereby mitigating the loss of information. Distill DSM achieves performance comparable to state-of-the-art 3D CNN model on BRATS 2020[2], pancreas[3], hippocampus[3] dataset and better results compared to 3D CNN on heart[3] and prostate[3] dataset with just **28%** parameters as compared to state of the art 3D CNN. Our paper has the following contributions-:

- We propose a distillation-based depth shift module that enables to segment volumetric data using 2D convolution by extracting and sharing necessary information to neighbouring slice along depth-dimension, reducing the model size to 28% of the state-of-the-art 3D CNN.

- The proposed solution is a plug-and-play module which could be incorporated with any 2D CNN architecture to model information along the Z direction.

- We did a comprehensive evaluation on five datasets to validate the proposed method.

## 2. Related Work

### 2.1. Segmentation

Earlier approaches to segment images use classical image processing techniques such as thresholding, region-growing methods, etc. (Vincent and Soille, 1991). Recent approaches

---

1. Originally Temporal shift Module (TSM), but since here volumetric data is used, it will be referred as Depth shift module (DSM) in further sections of paper

2. https://www.med.upenn.edu/cbica/brats2020/data.html

3. http://medicaldecathlon.com/

make use of machine learning techniques. Segmentation of 2D medical images using deep neural networks has an accuracy close to human performance today (Zhou et al., 2017; Shen et al., 2017; Falk et al., 2019; Ronneberger et al., 2015; Nandamuri et al., 2019). Initial approaches to segment 3D volumes used 2D CNNs to segment 2D slices individually (Milletari et al., 2016a). This approach, although being computationally friendly, it does not have good accuracy. More recently, fully convolutional architectures employ 3D convolutions such as 3D U-Net (Özgün Çiçek et al., 2016) and V-Net (Milletari et al., 2016b), which result in high performance but are computationally expensive.

### 2.2. Efficient Neural Network for learning 3D feature

Efficient neural network commonly uses 2D CNN along with some techniques to learn 3D feature in computationally inexpensive manner. Approaches for learning 3D feature using 2D CNN is mostly classified in three streams: 1) 2D slice distillation (Chen et al., 2016; Ettlinger et al., 2016; Cai et al., 2017; Novikov et al., 2019) 2) 2.5D (Prasoon et al., 2013; Ambellan et al., 2019; Li et al., 2018; Xia et al., 2018; Yu et al., 2019) and 3) 2D multiple views (Wang et al., 2019; Li et al., 2019). Methods adopting 2D slice distillation, distill 3D features from 2D features learned by 2D CNNs from 2D slices by employing recurrent neural network (rnn, 2001; Lipton et al., 2015) or conditional random field (Quattoni et al., 2005; Zheng et al., 2015). 2.5D based methods learns 3D features by giving several 2D slices as input to a 2D CNN. 2D multiple view based methods extract information from multiple views (usually: axial, coronal, and sagittal) and combine the information from multiple views for predicting the output. Another method generally adopted for making efficient CNN is to make use of binary kernel (Heinrich et al., 2018; Rastegari et al., 2016; Juefei-Xu et al., 2017). This approach reduces the parameter drastically.

Depth Shift Module (Lin et al., 2019) shifts part of feature channels in each frame to its neighbouring frame so that 2D convolution could handle depth information. Based on this idea of integrating depth information to 2D convolution, we propose Distill DSM.

## 3. Methodology

### 3.1. Problem Formulation

In the segmentation task for 3D image data, let $X_i$ and $Y_i$ represent input image volume and the segmentation maps respectively, where $X_i = \{x_1, x_2, ..., x_{N_i}\}$ and $Y_i = \{y_1, y_2, ..., y_{N_i}\}$, where $x_j \subseteq \mathbb{R}^{H \times W}$ is a 2D slice of medical image and $y_j$ is segmentation mask for the corresponding 2D slice. Different $X$ have different number of 2D slices i.e. different $N_i$. Our objective is to find $F$ such that objective function given below minimises.

$$J = \frac{1}{K} \sum_{i=1}^{i=K} L(F(X_i), Y_i) \tag{1}$$

where $K$ is total number of 3D volumes in the training dataset and $L$ is the loss function, which is computed using model output and ground truth.

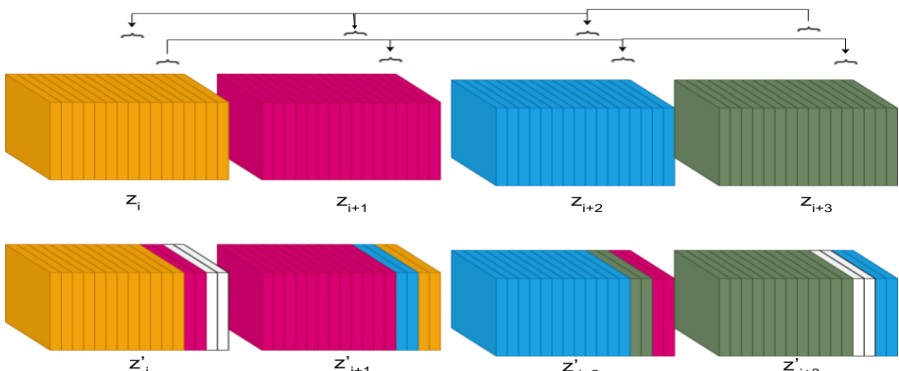

Figure 1: First row contains the intermediate convolutional features of the slices $\mathbf{Z_i}$, $\mathbf{Z_{i+1}}$, $\mathbf{Z_{i+2}}$, $\mathbf{Z_{i+3}}$. Some parts of the channels are shifted to neighbouring slices to exchange information. The second row contains convolutional features after shifting is done. The channel indicated by white color represents zero padding.

## 3.2. Intuition

3D CNN's capture inter slice information by convolving a 3D kernel to 3D input, which basically helps in gathering information from neighbouring slices to current slice. The operation results in gathering the complete information, including both spatial and semantic from the neighbouring slices. Using all the information from the neighbouring slices could be redundant in many cases. This makes them highly computationally and memory expensive.

A more efficient and simple way to exchange information among the neighbouring slices is to shift some channels of the current slice to neighbouring slices as shown in Figure 1 and proposed in (Lin et al., 2019). There have been various works (Bau et al., 2017, 2020) which show different channels correspond to different semantics. However, hard shifting of the channel will lead to loss of some information from the current slice including both semantic and spatial information. In order to prevent loss of information, (Lin et al., 2019) introduced residual DSM, where they add back the initial feature to channel shifted feature. However, addition is not an effective way to merge information and result in loss of some spatial and semantic information from current slice. This could lead to the drastic decrease in the performance of model, especially for segmentation task. If we can retain the necessary information (from the channels we are shifting) in the current slice and pass the necessary information required by neighbouring frame, then it would result in a highly efficient architecture having benefits of 3D architecture at a cost of 2D CNN. Motivated by this, we propose a novel architecture Distill Depth Shift Module(DSM).

## 3.3. Distill DSM

The proposed Distill DSM is shown in Figure 2. We extract three components of information from the part of feature channels which were shifted to neighbouring slices in DSM. Consider a feature map of $i^{th}$ frame $\mathbf{Z}_i \subseteq \mathbb{R}^{C \times h \times w}$ where $C$ is the number of channels and $h, w$ is the spatial size. We select $\alpha C$ channels from the end of $\mathbf{Z}_i$ where $\alpha \in [0, 1]$ and distill

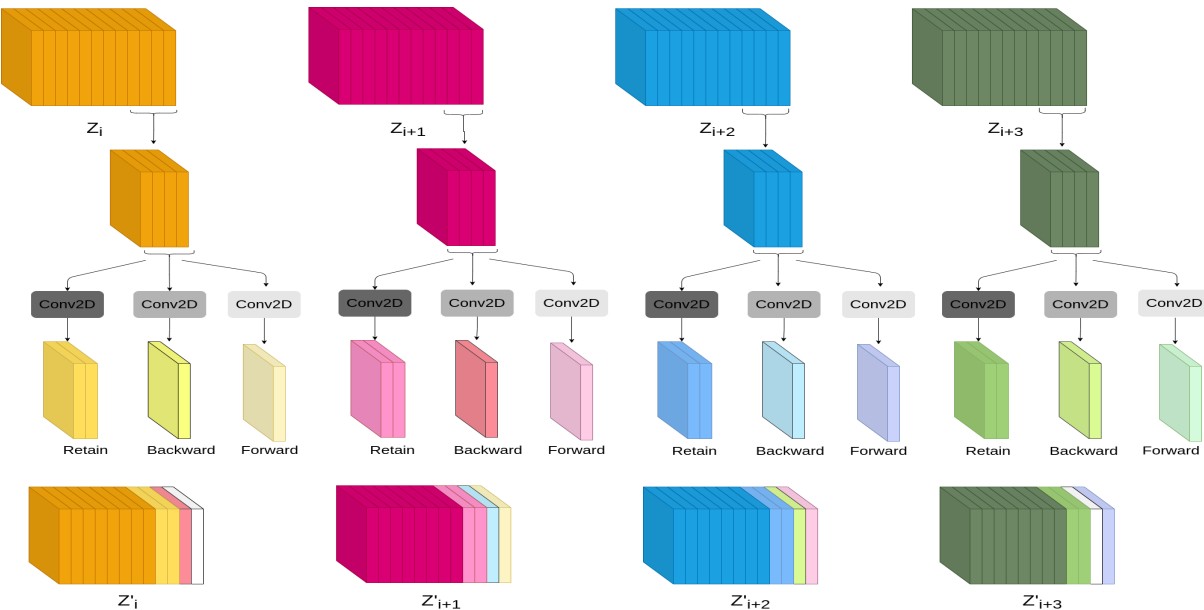

Figure 2: Above module represents Distill DSM, in which from a part of feature channels of every slice, three kinds of information are extracted. Features to retain, features to pass to forward slice, and features to pass to backward slice. Channels shown with white color in the second row represent zero-padded channels.

the information stored into three components as follows: 1) $\mathbf{R}_i \subseteq \mathbb{R}^{\alpha\frac{C}{2}\times h\times w}$: Necessary information to retain in $\mathbf{Z}_i$ 2) $\mathbf{F}_i \subseteq \mathbb{R}^{\alpha\frac{C}{4}\times h\times w}$: Necessary information to pass to forward slice $\mathbf{Z}_{i+1}$ 3) $\mathbf{B}_i \subseteq \mathbb{R}^{\alpha\frac{C}{4}\times h\times w}$: Necessary information to pass to backward slice $\mathbf{Z}_{i-1}$. In order to calculate the distilled information($\mathbf{R}_i, \mathbf{F}_i, \mathbf{B}_i$) we use a convolution layer for each of them. Now the retained information ($\mathbf{R}_i$) from current slice, forward information($\mathbf{F}_{i+1}$) and backward information($\mathbf{B}_{i-1}$) from the next and previous slice respectively are concatenated to the channels of current slice $\mathbf{Z}_i$. These operations are done for each and every slice in the volume. This completes the Distill DSM operation for the slices, after this each and every slice has information from previous and the following slice along with its own information. In the case of the first slice where there is no previous slice, the slice is zero padded in channel dimension to maintain the shape and similar adjustment is done for the last slice. The schematic for 4 slices is shown in Figure 2. Note, in the first Distill DSM layer only immediate neighbouring slices would share the information but as we go deeper in the network the slices which are far away would also be sharing the information0.

Using our approach for exchanging information helps in two ways. 1) Loss of information from the current slice is minimal as we perform distillation with the help of convolution to retain necessary information. 2) Information passed to forward and backward slice is not hard shifted i.e. model itself decides, what information it should pass and to what extent it should pass the information, as it can be the case in semantic segmentation that in initial layer the information exchange is more among neighbouring slices as to capture

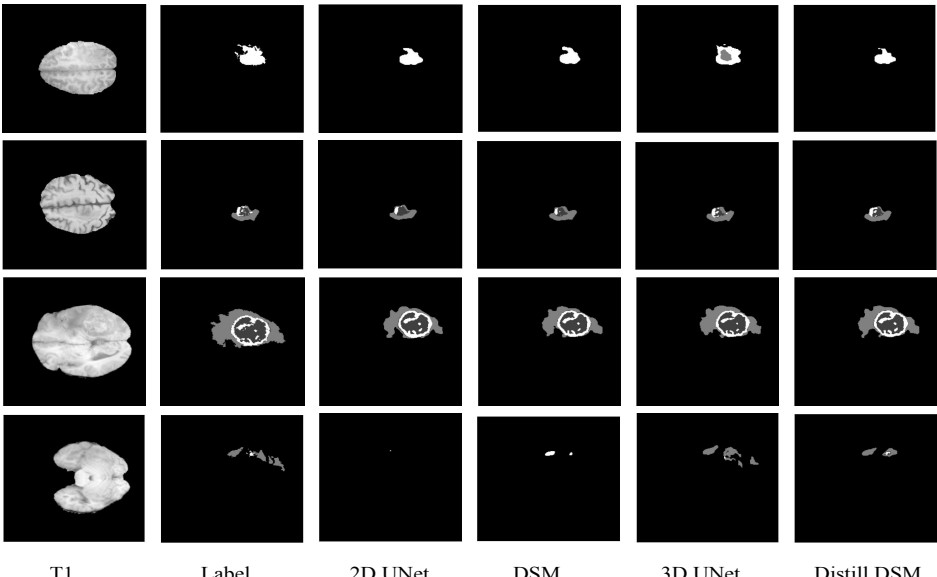

| T1 | Label | 2D UNet | DSM | 3D UNet | Distill DSM |

Figure 3: Results on BRATS 2020 dataset. First row, shows a case where 3D UNet fails to predict correctly where as Distill DSM predicts correctly. Last row is the boundary slice, where Distill DSM and 3D UNet are able to predict some of the segmentation maps where as DSM and 2D UNet are not, proving most essential thing that proposed model is able to model information along depth dimension

the spatial-temporal structures so that deeper layers could focus on semantics of current slice and hence segmenting the required portion.

## 4. Experiments

### 4.1. Datasets

To compare with the baseline (Ronneberger et al., 2015; Özgün Çiçek et al., 2016; Lin et al., 2019) we train and test our model on BRATS 2020 (Bakas et al., 2019) dataset and on 4 representative datasets of Medical Decathlon challenge (Simpson et al., 2019). The first dataset is Cardiac which includes 20 mono-modal MR volumes for segmentation of left atrium. The second is Hippocampus which includes 263 mono-modal MR volumes for segmentation of hippocampus head and body. The third is Prostate which includes 32 multi-modal MR volumes for segmentation of central gland and peripheral zone. The fourth being Pancreas, which includes 282 CT volumes for segmentation of liver and tumour. All the datasets from Medical decathlon were randomly splitted into 5 folds, by randomly shuffling the sequence of volumes and splitting the dataset into 5 fixed folds. Brats 2020 dataset consists of 371 training volumes and 127 testing volumes. Training data is further splitted into 4:1 ratio for training and validation and results are shown on testing dataset.

The Dice similarity coefficient and Hausdorff Distance 95 is used to evaluate proposed model for medical decathlon dataset and Dice similarity coefficient, Hausdorff Distance 95

Table 1: Quantitative segmentation results of 2D U-Net, 3D U-Net, Residual DSM and Distill DSM on BRATS 2020 dataset. ET represents Enhancing Tumor, WT represents Whole Tumor and TC represents Tumor Core

|  | Class | 2D U-Net | Residual DSM | 3D U-Net | Distill DSM(Ours) |
|---|---|---|---|---|---|
| Parameters | | 1,082,211 | 1,082,211 | 4,288,208 | 1,216,266 |
| Flops per voxel | | 38,662 | 38,735 | 58,709 | 39,456 |
| Wall time per voxel(s) | | 7.9498e-7 | 8.1726e-7 | 8.6517e-7 | 8.2641e-7 |
| Dice | ET | 0.712 | 0.732 | 0.704 | 0.753 |
| | WT | 0.861 | 0.867 | 0.879 | 0.873 |
| | TC | 0.687 | 0.704 | 0.796 | 0.742 |
| Sensitivity | ET | 0.714 | 0.707 | 0.687 | 0.761 |
| | WT | 0.859 | 0.835 | 0.898 | 0.841 |
| | TC | 0.660 | 0.693 | 0.779 | 0.726 |
| Specificity | ET | 0.9997 | 0.99978 | 0.99975 | 0.99969 |
| | WT | 0.99903 | 0.99939 | 0.99896 | 0.99944 |
| | TC | 0.99975 | 0.99986 | 0.99958 | 0.9997 |
| Hausdorff95 | ET | 35.20 | 29.21 | 43.27 | 30.52 |
| | WT | 6.52 | 8.42 | 11.46 | 5.98 |
| | TC | 27.39 | 34.85 | 18.84 | 32.87 |

Table 2: Ablation experiments for $\alpha$ hyper parameter

|  | Metric | $\alpha = \frac{1}{4}$ | $\alpha = \frac{1}{2}$ | $\alpha = 1$ |
|---|---|---|---|---|
| Parameter | | 1,115,606 | 1,216,266 | 1,619,330 |
| Heart | Dice 1 | 0.9125±0.008 | 0.9235±0.011 | 0.9203±0.009 |
| Hippocampus | Dice 1 | 0.8888±0.006 | 0.8955±0.005 | 0.8958±0.003 |
| | Dice 2 | 0.8618±0.006 | 0.8786±0.008 | 0.8795±0.002 |
| Prostate | Dice 1 | 0.8325±0.037 | 0.8724±0.014 | 0.8721±0.008 |
| | Dice 2 | 0.7565±0.065 | 0.7804±0.081 | 0.7818±0.076 |

(HSD), Sensitivity, and Specificity is used to evaluate proposed model on BRATS 2020 dataset.

## 4.2. Implementation details

Our experiments are implemented using PyTorch on NVIDIA Tesla V100 GPUs (16GB memory) and are carried out on Ubuntu machine with 96GB RAM and 32 cores. All networks use dice per channel loss function and Adam optimizer. Proposed distill DSM is integrated with U-Net (Ronneberger et al., 2015) architecture setting. After each convolutional operator in 2D U-Net, a distill DSM layer is added so that 2D convolution processes each slice individually and then pass it through Distill DSM to exchange information among the 2D slices. We have used $\alpha = \frac{1}{2}$ for experiments.

## 4.3. Ablation study

In this section value of $\alpha$ is determined. With increase in value of $\alpha$, the number of parameters increase and so is the information sent to forward and backward slice. Experiments were conducted on Heart, Hippocampus and Prostate dataset of medical decathlon challenge for varying value of $\alpha$ and is summarised in Table 2. It can be observed that from

Table 3: Quantitative segmentation results of 2D U-Net, 3D U-Net, Residual DSM and Distill DSM on heart, hippocampus, prostate and pancreas segmentation dataset from medical segmentation decathlon dataset

| Dataset | Metric | 2D U-Net | Residual DSM | VFN | 3D U-Net | Distill DSM(Ours) |
|---|---|---|---|---|---|---|
| Heart | Dice | 0.9025±0.004 | 0.9076±0.009 | 0.9085±0.010 | 0.918±0.009 | 0.9235±0.011 |
| | HD | 3.4723±0.118 | 1.8953±0.147 | 1.4588±0.153 | 1.2523±0.145 | 1.0056±0.123 |
| Hippocampus | Dice 1 | 0.8802±0.002 | 0.8901±0.007 | 0.8919±0.005 | 0.8993±0.004 | 0.8955±0.005 |
| | Dice 2 | 0.8618±0.011 | 0.8648±0.010 | 0.8663±0.009 | 0.8847±0.008 | 0.8786±0.008 |
| | HD | 1.6825±0.082 | 1.4209±0.045 | 1.4826±0.089 | 1.2587±0.075 | 1.3325±0.13 |
| Prostate | Dice 1 | 0.7847+0.041 | 0.7948±0.033 | 0.8068±0.052 | 0.8164±0.041 | 0.8724±0.014 |
| | Dice 2 | 0.6978±0.085 | 0.70214±0.07 | 0.7425±0.076 | 0.7339±0.066 | 0.7804±0.081 |
| | HD | 8.0664±0.567 | 6.8528±0.532 | 5.6835±0.692 | 5.5961±0.217 | 4.6294±0.485 |
| Pancreas | Dice 1 | 0.7395±0.024 | 0.7624±0.02 | 0.7650±0.018 | 0.7739±0.016 | 0.792±0.022 |
| | Dice 2 | 0.3485±0.036 | 0.3632±0.032 | 0.3684±0.031 | 0.4115±0.030 | 0.3765±0.035 |
| | HD | 16.7597±3.88 | 14.4535±3.82 | 13.6824±3.26 | 12.5574±1.04 | 11.9875±3.436 |

$\alpha = \frac{1}{4}$ to $\alpha = \frac{1}{2}$ there is huge improvement in performance, however from $\alpha = \frac{1}{2}$ to $\alpha = 1$ there isn't much improvement, but the parameters increased drastically. Hence, $\alpha = \frac{1}{2}$ is used for all the experiments as it results in parameter efficient and highly accurate model.

### 4.4. Comparison of results

In the comparison experiments, we compare proposed distill DSM with 2D segmentation method like U-Net (Ronneberger et al., 2015), 3D segmentation method like 3D U-Net (Özgün Çiçek et al., 2016) and computationally efficient method like residual DSM (Lin et al., 2019) and VFN (Xia et al., 2018). Table 1 summarises results on BRATS 2020 dataset (Bakas et al., 2019). Table 1 also have contains number of parameter, flops per voxel and wall time per voxel i.e. inference time per voxel. It can be observed that with very nominal parameter around 28% of 3D U-Net architecture we are able to achieve comparable results and our method outperform both residual DSM and 2D U-Net. Figure 3 visualises output of segmentation map from different methods.

Table 3 summarises results on various dataset of Medical Decathlon challenge (Simpson et al., 2019). Our method outperform 3D U-Net in case of heart and prostate dataset where we have limited number of 3D volume available for training. It is because of reason that 3D CNN are prone to over fitting especially when we have limited dataset available for training, making our method more efficient in such scenarios.

## 5. Conclusion

Our work focused on computationally efficient semantic segmentation module for volumetric data. We proposed a novel module Distill Depth shift module (Distill DSM) for efficiently using the information along the depth dimension with negligible increase in the parameters compared to 2D CNN. The proposed module can be inserted in any segmentation architecture to make use of depth information. We were able to achieve either better or comparable results to 3D CNN with only 28% of parameters. The proposed method was extensively tested on various datasets including BRATS 2020 and 4 datasets from medical decathlon challenge validating our the proposed method.

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
