# OpenReview forum: "Distill DSM: Computationally efficient method for segmentation of medical imaging volumes"
_MIDL.io/2021/Conference — MIDL 2021_

### Official Review · AnonReviewer4 · 2021-03-06

**Confidence:** 4
**Preliminary Rating:** 3
**Recommendation:** Oral
**Final Rating:** 4

**Summary:**

It is known that 3D CNNs can extract much needed 3D features for medical imaging segmentation, however they come with increased compute and memory usage. The authors propose a way for 2D convolutions to take into account neighbouring slices information, effectively introducing 3D information in 2D CNNs while using less parameters.


**Strengths:**

The paper is very well written and easy to follow.

I have seen works attempting to use neighbouring slices in the input (the so called 2.5D) but this is the first time that I see this idea applied to convolutions in medical imaging. Highlight should be given to the, as far as i know, novel idea of judging which features should be included on the forward and backward slice instead of simply shifting channels. It’s an interesting idea with potential for providing more efficient networks according to the authors findings.

Comparisons were made with famous architectures and a similar approach (Residual DSM) in multiple datasets, bringing more validity to the author’s claims.


**Weaknesses:**

Overall I do not see major weaknesses or problems in this manuscript. However, there are some minor problems.  Writing needs to be improved especially in the introduction, give attention to proper use of articles and verb tenses. Additionally, the introduction contains strong claims without proper citations. The explanation of how Distill DSM mixes information from far away slices could be improved (more minor details and suggestions in the detailed comments).

I see no promise to make code available, but that would make this work even stronger and easier to reproduce/be cited by future works.


**Deanonymize Review:**

no

**Detailed Comments:**

The introduction needs citations for your claims. Also, there are many strong affirmations that could be true for most but not all cases. An example in Section 1 Paragraph 3: “Using 2D CNNs for segmentation results in a [...] faster inference time”: this might not be true in all cases. Some 3D CNNs such as a 3D UNet can be extremely fast operating in inference mode (no gradient calculation), when able to fit the whole volume in memory for a one-shot prediction.

Writing could also be improved. Some examples of writing mishaps:

Section 1 Paragraph 3: “However it do not…”

Section 1 Paragraph 4: “In ordered to bridge…”

Section 2.1: “This approach, although being computationally friendly, it do not have good accuracy”.

One of the citations has “bak” as an author and author information is missing from References.

Correct me if i am wrong, but color coding in Figure 2 seems to be reversed, shouldn’t the “forward” features go to i+1? Instead they are shown going to i - 1, the same inversion is seen in “backward” features.

It was not clear to me how Distill DSM builds the volumetric segmentation. Does it follow the same typical approach of a 2D CNN, as in stacking slice predictions to form the volume? Section 3.3 should make this more clear.

In Section 3.3 a sentence caught my attention: “Note, in the first Distill DSM layer only immediate neighbouring slices would share the information but as we go deeper in the network the slices which are far away would also be sharing the information.”. This behaviour is crucial to understanding the proposed methodology and could be demonstrated better. Maybe show one more layer and the mixing of more than the immediate adjacent features. You could make room for this by making the Problem Formulation (Section 3.1) more succinct, since the authors did not use all the notation introduced there.


**Final Rating Justification:**

Considering the author's response, all of my concerns were addressed. As i said initially, i think the idea is very interesting and would be of interest to the community.

**Justification Of The Preliminary Rating:**

The methodology is solid and well validated. With some improvements in writing and referencing the rating could be improved. The paper presents an idea that would be of interest to the MIDL community.

**Paper Type:**

both

**Questions To Address In The Rebuttal:**

I would appreciate an answer to points given in the Weaknesses and Detailed comments section.

**Special Issue:**

no

---

> ### Author Response · Authors · 2021-03-17
> **Response to AnonReviewer4**
>
> We thank the reviewer for the comments. In regards to the statement “Using 2D CNNs for segmentation results in a ... faster inference time” refers to the comparison of models which have similar architectures, with the difference being in operators being 2D or 3D convolution. To substantiate our statement we have provided wall time (CPU execution time) per voxel in Table 1, Page 7 where it can be easily observed that 2D UNet wall time per voxel is lesser compared to 3D UNet. We have updated Figure 2, Page 5 thanks a lot for pointing it out.  In regards to information exchange among the slices, larger depth, we were unable to draw another diagram due to the limitation of the space and will try to answer the question here. Consider Figure 2, Page 5 after applying the  Distill DSM module we can see that $Z’_{i+1}$ contains the information of $Z_i$,  $Z_{i+1}$ and $Z_{i+2}$ and  $Z’_{i+2}$ contains the information of $Z_{i+1}$,  $Z_{i+2}$ and $Z_{i+3}$. When  $Z’_{i+1}$ will be passed through another Distill DSM module the information of $Z’_{i+1}$ will be passed to $Z’_{i+2}$. As $Z’_{i+1}$ contains the information of $Z_i$ now $Z’_{i+2}$ will also get some information from $Z_i$ which was absent in the previous Distill DSM module. Hence, we see as we apply Distill DSM module again and again the slices far away in the volume also start exchanging information. Distill DSM builds the volumetric segmentation, by taking in a 3D input volume which will have 2D multiple slices. Then for each convolutional operation in Encoder and Decoder, it processes each slice individually using 2D convolution and then passes it through Distill DSM to exchange information among the 2D slices. We have also added a description of the same in Section 4.2, Page 7. In regards to the citation, we have corrected that.

---

### Official Review · AnonReviewer2 · 2021-03-08

**Confidence:** 4
**Preliminary Rating:** 3
**Recommendation:** Poster
**Final Rating:** 3

**Summary:**

The authors proposed a computationally efficient semantic segmentation module Distill Depth shift module (Distill DSM) for volumetric data.  It is designed to enable 2D convolutions in each layer of the network to extract information from a part of the channels and share it with neighboring slices, thus facilitating information exchange among neighboring slices with introducing few extra learn-able parameters. The authors did a comprehensive evaluation based on BRATS 2020 and 4 datasets from medical decathlon challenge to validate its effectiveness.


**Strengths:**

1.  The paper is written clearly. The description of the method and the experiments are reasonable. I can easily understand the proposed module.
2.  The proposed Distill DSM is easy to follow and can be plugged into different 2D networks for 3D volumetric segmentation.
3.  The authors conducted abundant and well-organized experiments to validate the effectiveness of proposed Distill DSM.

**Weaknesses:**

1. The proposed Distill DSM lacks theory or reference supporting. It seems like an engineering application rather than innovation in method. If the author would like to prove the novelty of their proposed method, it needs more theoretical explanation and method reference.

2. The description and visualization of methods seems  to be too simple, which easily make readers confused. For instance, the description “Depth Shift Module (Lin et al., 2019) shifts part of feature channels in each frame to its neighbouring frame so that 2D convolution could handle depth information.”, does the DSM shifts the same features in each frame to its last and next frames? What the Fig. 1 presents is not the same as the description.

**Deanonymize Review:**

no

**Detailed Comments:**

1.  Fig. 1 provides a rough description, and some details need to be properly illustrated. For instance, what is the difference between the two horizontal lines with arrows? And how many channels are shift to the neighboring frames?
2. In general the visualization of method (Fig. 1 and Fig. 2) have low quality and they must be improved.
3. Some discussions on applying the DSM modules to other network structures would be helpful.


**Final Rating Justification:**

The authors clarified the motivation. Some minor suggestions in the original comment are expected to be considered in the final version.

**Justification Of The Preliminary Rating:**

The proposed method was overall novel and the authors validated the method with serval datasets. The results seems to be convincing. This paper is overall well written and easy to follow. I would recommend acceptance.

**Paper Type:**

methodological development

**Questions To Address In The Rebuttal:**

1.  The authors should give more theoretical explanation or method references to support the proposed Distill DSM.
2.  In the experiments, the inference time need to be reported since the authors aimed to achieve high computational efficiency.


**Special Issue:**

no

---

> ### Author Response · Authors · 2021-03-17
> **Response to AnonReviewer2**
>
> We thank the reviewer for the insights. The innovation in our model is using distillation in the DSM module [1] such that the information exchange between the slices happens without the loss of information. The theoretical explanation of the proposed approach is explained in Section 3.3, Page 5 and in Figure 2, Page 5. In order to provide a better insight into the computational efficiency of the proposed approach, we have provided wall time (CPU execution time) per voxel and FLOPS per voxel in Table 1, Page 7. Figure 1 is used to provide insight and visual representation of how actually depth shift module works, the channel shifted are not on the scale. Discussion about the number of channels to shift is provided in section 3.3, Page 5 and section 4.3, Page 7. Further, In Figure 1, Page 4 the horizontal line represents the shifting of channels. The top horizontal line represents channels shifting backwards and the bottom horizontal line represent the channels shifting forward.
>
> [1] Ji Lin, Chuang Gan, and Song Han. Temporal shift module for efficient video understanding. arXiv preprint arXiv:1811.08383, 2018.

---

### Official Review · ~Hongwei_Bran_Li1 · 2021-03-08

**Confidence:** 4
**Preliminary Rating:** 3
**Recommendation:** Poster
**Final Rating:** 3

**Summary:**

The authors proposed to enable 2D convolutions to make use of information from neighboring by a Distill DSM which learns to extract information from a part of the channels and shares it with neighboring slices.  The model is extensively evaluated on several datasets. The proposed model achieves better performance than 3D CNN for heart and prostate datasets and comparable performance on BRATS 2020, pancreas, and hippocampus dataset with simply 28% of parameters compared to 3D CNN model.

**Strengths:**

The paper is well written. Two main strengths in my opinions:
1. Enabling 2D convolutions to make use of information from neighboring by a Distill DSM which learns to extract information from a part of the channels and shares it with neighboring slices.
2. Extensive evaluations on several datasets. The proposed model achieves better performance than 3D CNN for heart and prostate datasets and comparable performance on BRATS 2020, pancreas, and hippocampus dataset with simply 28% of parameters compared to 3D CNN model.

**Weaknesses:**

1. Another important class of efficient 2D&3D approaches is not mentioned and discussed.

2. Some explanation of the results on BraTS 2020 and some clarification are needed. For example, the evaluation metric called specificity for BraTS 2020 may not be needed.



**Deanonymize Review:**

yes

**Detailed Comments:**

The author presents an interesting efficient approach to handle 3D segmentation tasks by incorporating 2D convolution layer with a Distill DSM which learns to extract information from a part of the channels and shares it with its neighboring slices.
There are some concerns raised when going through the paper.

1. One important class of efficient CNN, which makes use of fixed/binary kernels [1, 2, 3] and drastically reduces learnable parameters are not mentioned or compared. They even need fewer parameters.

2. Although the methods mentioned above are not compared, the authors present extensive experiments with 3D-UNet and other methods with an ablation study.

3. The evaluation metric 'Specicity' seems not appropriate as it considers more on the background pixels and is very closed to 1. Why Distill DSM is poor on the class Tumor Core (TC) on BraTS. Do the authors have any idea?

4. Does the number of slices adaptive in the experiments? I see alpha is set to 0.5 but I assume it depends on the structure of pathology or organs? Please justify this.




[1] Obelisk-one kernel to solve nearly everything: Unified 3d binary convolutions for image analysis.
[2] Xnor-net: Imagenet classification using binary convolutional neural networks.
[3] Local binary convolutional neural networks.


**Final Rating Justification:**

Thanks for the response.
Almost all my of concerns were addressed.

**Justification Of The Preliminary Rating:**

1. well-written and an interesting approach.

2. Another important class of efficient 2D&3D approaches is not mentioned and discussed.

3. Some explanation of the results on BraTS 2020 and some clarification are needed. For example, the evaluation metric called specificity for BraTS 2020 may not be needed.


**Paper Type:**

methodological development

**Questions To Address In The Rebuttal:**

Mostly from the weakness part.

1. One important class of efficient CNN, which makes use of fixed/binary kernels [1, 2, 3] and drastically reduces learnable parameters are not mentioned or compared. They even need fewer parameters.

2. The evaluation metric 'Specicity' seems not appropriate as it considers more on the background pixels and is very closed to 1. Why Distill DSM is poor on the class Tumor Core (TC) on BraTS. Do the authors have any idea?

3. Does the number of slices adaptive in the experiments? I see alpha is set to 0.5 but I assume it depends on the structure of pathology or organs? Please classify this.


**Special Issue:**

no

---

> ### Author Response · Authors · 2021-03-17
> **Response to AnonReviewer1**
>
> We thank the reviewer for the insightful comments. In regards to comparison with binary kernel CNN, we have added a description of the approach in relevant work (Section 2.2, Page 3) to give better insight. Also, the proposed approach effectiveness could be easily validated through current experiments. We have added a comparison with the 2.5D approach VFN[1] in Table 3, Page 8 to provide more insight and evidence that the proposed approach outperforms other computationally efficient methods. In regards to mentioning “specificity” as an evaluation metric, we have done that to maintain consistency as BRATS evaluation portal (https://ipp.cbica.upenn.edu/) also provides them. The drop in accuracy of class Tumor Core (TC) for BRATS dataset can be attributed to the trade-off between the model capacity and its performance. It may be noted that we have also achieved an improvement of 5% on class Enhancing Tumor (ET). Also to provide better insight into the hyperparameter $\alpha$ and it’s functioning, we have performed the ablation study on few more datasets and summarised the result in Table 2, Page 7.
>
> [1] Yingda Xia, Lingxi Xie, Fengze Liu, Zhuotun Zhu, Elliot K. Fishman, and Alan L. Yuille. “Bridging the gap between 2d and 3d organ segmentation with volumetric fusion net”, International Conference on Medical Image Computing and Computer-Assisted
> Intervention (2018) 445–453

---

### Official Review · AnonReviewer3 · 2021-03-09

**Confidence:** 5
**Preliminary Rating:** 3
**Recommendation:** Poster

**Summary:**

The authors propose a fast and versatile segmentation method for image volumes called a distillation-based depth-shift module. The main approach is motivated by the work of Lin et al. The main novelty is the introduction of a distillation-based approach that leads to sizable gains in accuracy compared to both 2D and 3D UNet (depending on the dataset).

**Strengths:**

The idea of learning feature maps for spatially adjacent slices is simple and general.

The reported results are very competitive and were achieved on public datasets.

The manuscript is clearly written and easy to understand.

The contributions of the paper are clear and substantiated.



**Weaknesses:**

The method could be compared to more segmentation variants, e.g., 2.5D segmentation approaches.

Utilizing feature maps from neighboring slices is not entirely novel. A similar idea has been previously suggested for videos by Lin et al. (2019).



**Deanonymize Review:**

no

**Detailed Comments:**

Could the font size in Figure 1 be increased?

The word "retainInfo" (p. 5) is written in italic font. It seems to refer to the variable but is also used as an operator further down in the same paragraph. It would be great to clarify this.

Please cite the original work of an idea, e.g., 2.5D have been suggested by Prasoon et al. in 2013, not by Li et al. (who themselves have cited this work correctly).


**Justification Of The Preliminary Rating:**

The paper is clearly written and evaluated on several public datasets. The authors provide some technical novelty in their approach but still have adopted the underlying idea from already published work.

**Paper Type:**

validation/application paper

**Questions To Address In The Rebuttal:**

How does the model compare to 2.5D approaches?

**Special Issue:**

no

---

> ### Author Response · Authors · 2021-03-17
> **Response to AnonReviewer3**
>
> We thank the reviewer for summarizing the paper and providing insightful comments. We have increased the font size in Figure 1 in the modified manuscript as per suggestion. In regards to the term “retainInfo” we have made suitable changes in the new manuscript to remove the confusion in Section 3.3, Page 5. We have removed the term “retainInfo” and added a description to make it clear. We have cited the paper that originally provided the idea for 2.5 approaches in the new manuscript. In order to have a comparison with 2.5D approaches, we have trained Volumetric Fusion Net [1] with the Medical decathlon dataset and summarised the result in Table 3, Page 8.
>
> [1] Yingda Xia, Lingxi Xie, Fengze Liu, Zhuotun Zhu, Elliot K. Fishman, and Alan L. Yuille. “Bridging the gap between 2d and 3d organ segmentation with volumetric fusion net”, International Conference on Medical Image Computing and Computer-Assisted
> Intervention (2018) 445–453

---

### Meta-Review · Area_Chair1 · 2021-03-29

**Recommendation:** Accept (Poster)

**Metareview:**

The reviewers find the work of interest and there was initial consensus that the paper can be accepted. This was confirmed after the rebuttal.

**Paper Type:**

both

---

### Decision · Program_Chairs · 2021-03-31

Accept